# Analysis of the Partial Nitrification/Anammox Performance and Microbial Structure of Low C/N Wastewater by A²/O Process

**Lei Ye, Yanhao Zhou, Liangdong Tang, Sixing Chen and Xianguang Zhao \***

College of Environmental Science and Engineering, Nanjing Tech University, Nanjing 211816, China; 202061102005@njtech.edu.cn (L.Y.); 202161202045@njtech.edu.cn (Y.Z.); 202161202027@njtech.edu.cn (L.T.); 202161202015@njtech.edu.cn (S.C.)
**\*** Correspondence: zxg174@163.com

**Abstract:** Given the carbon limitation of low C/N wastewater, the improvement of nitrogen-removal efficiency remains a challenging task of municipal wastewater treatment plants (WWTPs) in China. In this study, a partial nitrification/anammox (PN/A) system was established to facilitate the anaerobic-anoxic-aerobic (A²/O) treatment of low C/N (C/N = 3) wastewater with insufficient carbon sources. Effects of dissolved oxygen (DO) concentration and internal reflux ratio on nitrogen-removal efficiency and pathway were investigated. Under the optimal DO (0.5–0.8 mg·L$^{-1}$) and internal reflux ratio (250%), the highly efficient $NH_4^+$-N removal (97.21%) and TN removal (80.92%) were achieved based on PN/A. Moreover, the relative abundance of ammonia-oxidizing bacteria (*Nitrosomonas*) was 3 times higher than the abundance of nitrite-oxidizing bacteria (*Nitrospira*) in phase V, which was the main cause of PN in the reactor. Anaerobic ammonia-oxidizing bacteria (*Candidatus Brocadia*, *Pirellula*, and *Gemmata*) were also found and considered as the key microbes involved in anammox. This study reports that the A²/O process can achieve advanced nitrogen removal of low C/N wastewater based on PN/A by optimizing conventional process parameters. The outcomes of this study may provide practical engineering applications as a reference for nitrogen removal based on the A²/O process.

**Keywords:** anaerobic-anoxic-aerobic process; wastewater treatment; partial nitrification; anammox; microbial community structure





## 1. Introduction

Sewage discharge in natural water bodies has become a serious environmental concern, as it contains large amounts of nitrogenous pollutants that cause eutrophication of water bodies [1]. Water pollution causes serious health concerns and limits the availability of clean water for humans as well as for the ecosystem. Therefore, it is necessary to conduct efficient and stable nitrogen-removal treatment of sewage. In order to remove the pollutants in wastewater, various degradation technologies such as the physicochemical method [2], photocatalytic technology [3], and biodegradation [4] method are recommended. Currently, the traditional biological nitrogen-removal (BNR) method is most commonly used for wastewater treatment, which is facilitated by nitrification and denitrification processes [5]. However, the traditional BNR technology is less efficient for treating high $NH_4^+$-N and low C/N wastewater. Due to insufficient influent carbon sources in wastewater treatment plants (WWTPs) in China, additional organic carbon sources need to be added during the denitrification stage to improve the removal efficiency [6]. Additionally, low nitrogen removal efficiency caused by insufficient feedwater carbon sources is also a bottleneck of BNR. As the water quality standards of WWTPs in China become more stringent, it is of great significance to upgrade the sewage treatment plants to improve the nitrogen removal efficiency of BNR system.

Partial nitrification/anammox (PN/A), a new method of nitrogen removal, is considered a promising alternative to BNR due to its autotrophic denitrification [7]. In the

PN/A process, approximately 50% of $NH_4^+$-N is converted to $NO_2^-$-N by the oxidation of ammonia-oxidizing bacteria (AOB) (Equation (1)) [8]. Meanwhile, ammonia-oxidizing bacteria (AnAOB) use $NO_2^-$-N generated as an electron acceptor to convert $NH_4^+$-N remaining to $N_2$, according to Equation (2) [9].

$$NH_3 + 1.5\ O_2 \rightarrow NO_2^- \text{-N} + H_2O + H^+ \tag{1}$$

$$NH_3 + 1.32\ NO_2^- + H^+ \rightarrow 1.02\ N_2 + 2\ H_2O + 0.26\ NO_3^- \tag{2}$$

Therefore, the PN/A process offers unique advantages of no external carbon sources consumption, high nitrogen removal efficiency, and low DO demand [10,11]. The PN/A process has become a research hotspot for the treatment of low C/N wastewater [12]. By regulating the hydraulic retention time (HRT), Hou et al. [13] initiated anammox in a sequencing batch reactor (SBR) and an up-flow anaerobic sludge bed reactor (UASB) at ~25 °C and <0.5 mg·$L^{-1}$ DO. Wu et al. [14] achieved enrichment of ammonia-oxidizing bacteria (AOB) and anaerobic ammonia-oxidizing bacteria (AnAOB) while maintaining the reflux ratio at 300%. Through real-time monitoring and control of pH, Yang et al. [15] initiated PN/A in an SBR reactor. These studies showed that a stable PN/A system could be achieved by optimizing the parameters for increasing nitrogen removal efficiency and reducing the requirement for additional carbon sources. However, in most of these studies, an appropriate amount of AnAOB was added to the reactors. At the same time, the reactors were intentionally fed with low organic matter concentration and sufficient $NO_2^-$-N influent water, which promoted the metabolism and reproduction of AnAOB. These are difficult to achieve in the practical application of sewage treatment engineering. Few researchers have investigated PN/A through anaerobic-anoxic-aerobic ($A^2$/O) process, which is the most mainstream wastewater treatment process in domestic WWTPs. Approximately 21% of total wastewater is treated by these WWTPs using $A^2$/O [16]. Simultaneously, the $A^2$/O process has the advantages of low operating cost, strong impact resistance, easy maintenance, and high adjustability, which make it conducive for engineering applications. Therefore, it is important to improve the denitrification efficiency of $A^2$/O processes during low C/N wastewater treatment by adjusting conventional process parameters.

In this study, a PN/A-based $A^2$/O process was developed to treat wastewater with low C/N. Changes in the nitrogen-removal ability, pollutant transformation, and microbial community structure were investigated in response to the varying operational parameters (i.e., DO and reflux ratio). The study aims to provide new ideas for the PN/A application in WWTPs.

## 2. Materials and Methods

### 2.1. Operation of the Continuous Process

The experimental system consisted of an $A^2$/O reactor (working volume: 13.8 L) and a secondary sedimentation tank (5 L) (Figure 1). As shown in Figure 1, the $A^2$/O reactor was composed of six compartments of the same size, including one cell in the anaerobic zone, two cells in the anoxic zones, and three cells in the aerobic zones. The synthetic wastewater was pushed upward, passed through each cell successively, and finally flowed into the secondary sedimentation tank. The aerobic sections were oxygenated by a microporous aeration system, and the DO concentration was controlled by a gas rotor flowmeter timely. The inflow rate, nitrification reflux ratio, and sludge reflux ratio of the reaction system were adjusted by peristaltic pumps. Mechanical stirrers were set up in the anaerobic and anoxic zones to supply for mixing.

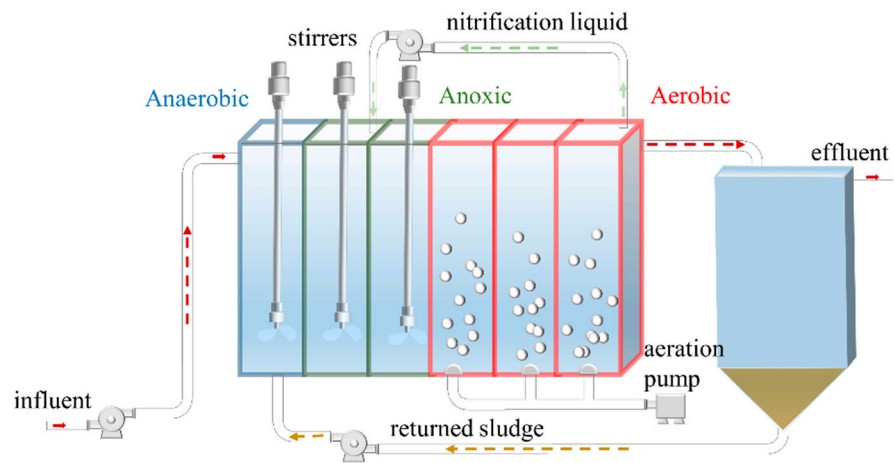

**Figure 1.** Schematic diagram of the experimental set-up.

During the operating stage, HRT and sludge retention time (SRT) were maintained at 14 h and 15–20 days, respectively. The concentration of mixed liquid-suspended solids (MLSS) was about 3100–3460 mg·L$^{-1}$. The temperature was controlled at around 25 °C using heaters. Synthetic wastewater with CH$_3$COONa and NH$_4$Cl (C/N = 3) was used as the feed. According to the regulation of the DO concentration of the aerobic zones and nitrification reflux ratio in the A$^2$/O process, the experimental process was divided into five phases. The operating parameters and characteristics of the influent are shown in Table 1.

**Table 1.** Operating parameters and influent characteristics during different periods.

| Items | Phase I | Phase II | Phase III | Phase IV | Phase V |
|---|---|---|---|---|---|
| Days/d | 1–30 | 31–60 | 61–90 | 91–111 | 112–132 |
| DO/mg·L$^{-1}$ | 2.0–3.0 | 1.0–1.5 | 0.5–0.8 | 0.5–0.8 | 0.5–0.8 |
| Nitrification reflux ratio (R1)/% | 200 | 200 | 200 | 250 | 300 |
| Sludge reflux Ratio (R2)/% | 100 | 100 | 100 | 100 | 100 |
| Inf.COD/mg·L$^{-1}$ | 231.36 ± 0.78 | 229.35 ± 4.51 | 230.36 ± 3.02 | 229.84 ± 1.52 | 228.54 ± 2.11 |
| Inf.NH$_4^+$-N/mg·L$^{-1}$ | 77.01 ± 1.95 | 77.16 ± 1.53 | 77.62 ± 4.68 | 77.17 ± 2.14 | 76.57 ± 0.88 |
| Inf.NO$_2^-$-N/mg·L$^{-1}$ | 0.65 ± 0.08 | 0.71 ± 0.05 | 0.72 ± 0.13 | 0.84 ± 0.07 | 0.73 ± 0.11 |
| Inf.NO$_3^-$-N/mg·L$^{-1}$ | 0.83 ± 0.04 | 0.82 ± 0.14 | 0.91 ± 0.06 | 0.83 ± 0.15 | 0.82 ± 0.07 |
| Inf.pH | 8.19 ± 0.15 | 7.96 ± 0.14 | 8.07 ± 0.68 | 7.87 ± 0.23 | 7.96 ± 0.08 |

*2.2. Analytical Methods*

The concentrations of NH$_4^+$-N, NO$_3^-$-N, NO$_2^-$-N, total nitrogen (TN), and MLSS were analyzed following the standard method recommended by the Chinese State Environmental Protection Administration, 2002. COD was measured through HACH fast digestion method. The pH and temperature were monitored by Leici PHS-25 pH meters. DO was measured by Leici JPB-607A DO probes.

*2.3. Calculations of Experimental Index*

2.3.1. Calculation of Nitrite Accumulation Rate

Calculation method of nitrite accumulation rate (NAR):

$$NAR = \frac{NO_2^- - N}{NO_2^- - N + NO_3^- - N} \tag{3}$$

where, NO$_2^-$-N and NO$_3^-$-N denote the concentrations of NO$_2^-$-N and NO$_3^-$-N in the last aerobic cell, respectively.

2.3.2. Calculation of Pollutants Mass Balance

The mass balance of the nitrogenous pollutants is shown in Figure 2. The nitrogenous pollutant material balance in the anaerobic and anoxic zones in the system could be calculated as follows using Equations (4) and (5).

$$\Delta C_{ana} = \frac{Q_{inf} \times C_{inf} + Q_{slu} \times C_{eff}}{Q_{inf} + Q_{slu}} - C_{ana} \tag{4}$$

$$\Delta C_{ano} = \frac{C_{ana} \times \left(Q_{inf} + Q_{slu}\right) + Q_R \times C_{aer}}{Q_{inf} + Q_{slu} + Q_R} - C_{ano} \tag{5}$$

where, $\Delta C_{ana}$ and $\Delta C_{ano}$ represent the concentrations of pollutants changed in the last cells of the anaerobic zone and anoxic zones (mg·L$^{-1}$), respectively. $C_{inf}$ and $C_{eff}$ denote pollutants concentrations of influent and effluent (mg·L$^{-1}$), respectively. $C_{ana}$, $C_{ano}$, and $C_{aer}$ are the concentrations of pollutants at the end of the anaerobic zone, anoxic zones, and aerobic zones (mg·L$^{-1}$), respectively. $Q_{inf}$, $Q_{eff}$, $Q_R$, and $Q_{slu}$ are inlet and outlet water flow, internal return flow, and sludge return flow (L·h$^{-1}$).

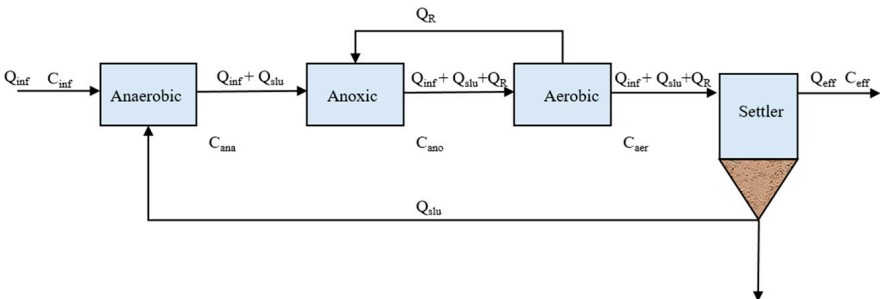

**Figure 2.** Material balance of pollutants.

*2.4. Microbiological Analysis*

DNA extraction, gene amplification, and high-throughput sequencing were used to analyze the composition of microorganisms in the reactor. Sludge mixture samples were collected from each reaction zone (anaerobic, anoxic, and aerobic) on the last day of the test phase. In this test, 5 mL mixed sample was taken and centrifuged at 10,000 r/min for 10 min. The supernatant was discarded, and DNA was extracted following the instruction of the Fast DNA Spin Kit for Soil. All samples were stored at −80 °C. The sample was checked for the quality of the extracted DNA by agarose gel electrophoresis, and we used a spectrophotometer to determine DNA concentration and purity.

Primer sequences 515F (5′-GTGCCAGCMGCCGCGGTAA-3′) and 806R (5′-GGACTAC HVGGGTWTCTAAT-3′) in the V4 region of the bacterial 16S rRNA were selected for PCR amplification. High-throughput sequencing was then performed on the Illumina MiSeq platform. The effective sequences obtained after sample homogenization were classified using the Ribosomal Database Project (RDP), and the microbial community characteristics of the system were analyzed simultaneously.

**3. Results and Discussion**

*3.1. Performance of Nitrogen Removal under Different DO Concentrations*

NH$_4^+$-N and TN removal as well as concentrations in influent and effluent under different DO concentrations (phases I–III), are shown in Figure 3. The average concentrations of NH$_4^+$-N in the effluent during phase I, II, and III were 1.96, 3.21, and 2.39 mg·L$^{-1}$, respectively (Figure 3a). The observed NH$_4^+$-N concentrations in the effluent were less than the pollutant discharge standards set for the municipal wastewater treatment plants (GB18918-2002) [17] in China (NH$_4^+$-N ≤ 5.0 mg·L$^{-1}$). In phase I (DO in the aerobic zones:

2–3 mg·L$^{-1}$), enrichment of nitrifying bacteria was observed, and the nitrification effect was the best. Therefore, NH$_4^+$-N may have been mostly converted into NO$_3^-$-N, and NO$_2^-$-N did not accumulate. With the decrease in DO concentration during phase II and III, the NH$_4^+$-N concentrations in effluent varied. However, the average NH$_4^+$-N removal efficiencies remained above 95%. Due to sufficient HRT (6.9 h) in the aerobic zone, oxidation of NH$_4^+$-N was less affected by the decrease in DO concentration. Liu et al. [18] also reported good nitrification in the system with low DO concentration (0.37 mg·L$^{-1}$), sludge age of 10 days, and HRT of 6 h in the aerobic zone.

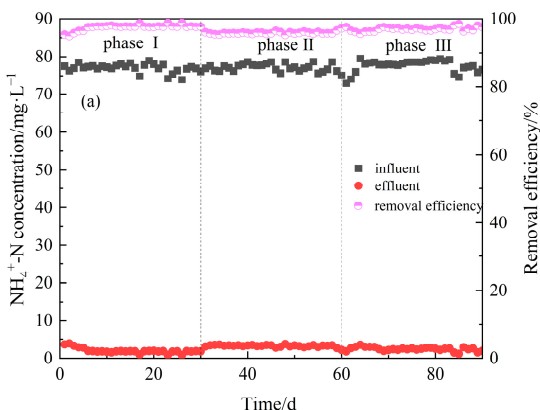 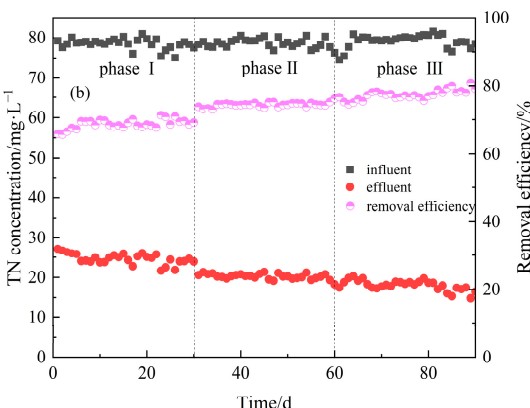

**Figure 3.** Impact of the DO concentration on (**a**) NH$_4^+$-N and (**b**) TN removal efficiencies.

Figure 3b shows that the TN concentrations in the effluent during phases I, II, and III were 24.61, 20.22, and 18.03 mg·L$^{-1}$, respectively, while the TN removal efficiencies were 68.65%, 74.29%, and 77.23%, respectively. From phase I to phase III, the DO concentrations were 0.13, 0.09, and 0.05 mg·L$^{-1}$ in the anaerobic zone, while 0.26, 0.15, and 0.10 mg·L$^{-1}$ in the anoxic zone. The DO concentration in the reflux nitrification solution was higher in phase I, which could decrease the denitrification efficiency. In phases II and III, the DO concentration in the aerobic zone continued to decrease, while the anoxic zone remained in an anaerobic state with 0.1 mg·L$^{-1}$ DO, which resulted in enhanced denitrification and nitrogen-removal efficiency. This finding was in agreement with the previous research reported by Li et al. [19]. Li et al. observed that a decrease in DO concentration in the aerobic zone (from 2 to 0.5 mg·L$^{-1}$) led to improved denitrification and a subsequent increase in TN removal rate.

Three parallel experiments were conducted to detect the concentrations of NH$_4^+$-N, NO$_3^-$-N, and NO$_2^-$-N in the different sections of the reactor during phases I to III. At the same reflux dilution (200%), good NH$_4^+$-N removal was observed due to nitrification in the aerobic zone. However, NH$_4^+$-N removal was significantly different in the anaerobic and anoxic zones. The changes in NH$_4^+$-N concentration in the system during phase I-III are shown in Figure 4a. The removal rates of NH$_4^+$-N in the anaerobic and anoxic zones during phase I, II, and III were 5.22%, 9.21%, and 14.12%, respectively. The gradually enhanced NH$_4^+$-N removal efficiency in the anaerobic and anoxic zones indicated the existence of certain anammox denitrification phenomena in the anaerobic and anoxic zones in the A$^2$/O reactor.

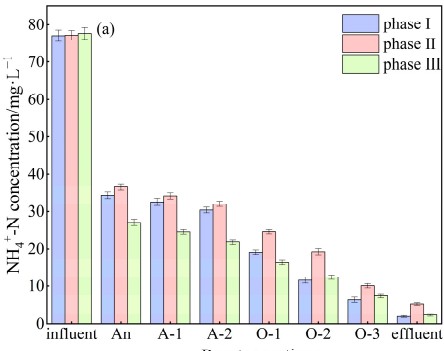 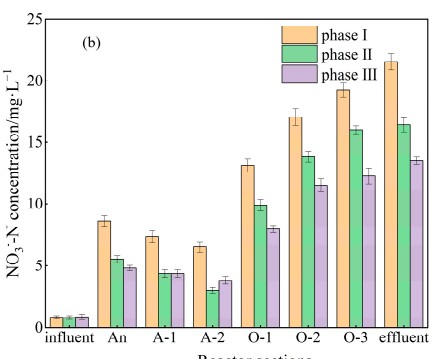 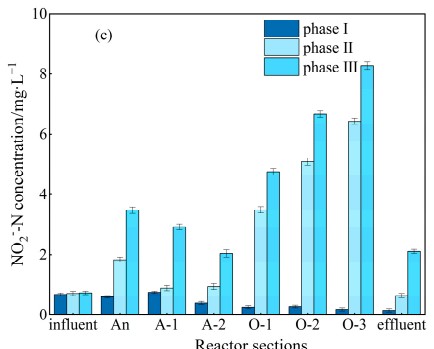

**Figure 4.** (**a**) $NH_4^+$-N, (**b**) $NO_3^-$-N, and (**c**) $NO_2^-$-N concentrations in the different sections of the reactor.

As shown in Figure 4b, no noteworthy difference was observed in the removal of $NO_3^-$-N in the anoxic zone, while the generation of $NO_3^-$-N in the aerobic zone began to show an obvious decreasing trend. Meanwhile, in phase II and III, the $NO_2^-$-N concentration in the aerobic zone increased significantly (Figure 4c). This increase can be attributed to the partial nitrification of $NH_4^+$-N in the aerobic zone with low DO concentration. The accumulation rates of nitrite nitrogen (NAR) in the last aerobic cell during phase I, II, and III were 0.93%, 28.68%, and 40.24%, respectively. NAR of the last aerobic cell increased substantially in phase II, and partial nitrification occurred in the aerobic zone in phase III. This finding was consistent with the research by Wei et al. [20], in which partial nitrification was initiated by controlling the DO concentration between 0.5 to 0.8 mg·L$^{-1}$ at around 25 °C.

AOB shows stronger hypoxic affinity and doubling rate than nitrate-oxidizing bacteria (NOB). By controlling DO concentration in the reactor, selective amplification of AOB and elimination of NOB can be achieved progressively [21,22]. $NO_2^-$-N accumulated due to partial nitrification flowed back to the anaerobic and anoxic zones and was rapidly utilized by AnAOB.

### 3.2. Performance of Nitrogen Removal under Different Reflux Ratios

At the beginning of the experiment, an internal reflux ratio of 200% was maintained. During the optimization tests (phase IV–V), the pollutant removal efficiency of the process was investigated. Reflux ratios of nitrification liquid in phases IV and V were 250% and 300%, respectively. Figure 5a illustrates the effect of three different reflux ratios on $NH_4^+$-N removal efficiency. For reflux ratios of 200%, 250%, and 300%, the average $NH_4^+$-N removal efficiencies were 96.92%, 97.21%, and 97.33%, respectively, and effluent concentrations were less than 2.39, 2.15, and 2.07 mg·L$^{-1}$, respectively. These observations suggest that the reflux ratio did not have any remarkable impact on the $NH_4^+$-N removal efficiency. During phases IV and V, $NH_4^+$-N concentrations in effluent were higher than that of phase III. This increase can be attributed to the increase in reflux ratio during phases IV and V, which shortened the HRT in the aerobic zone and inhibited nitrification.

The average TN concentrations in the effluent from phase III to phase V were 18.03, 14.93 and 13.90 mg·L$^{-1}$, respectively, while the respective TN removal efficiencies (average) were 77.24%, 80.92%, and 81.13% (Figure 5b). At reflux ratios of 250% and 300%, the effluent quality satisfied the first-class discharge A criteria (GB 18918-2002). With the increase in nitrification reflux ratio, the TN removal rate increased, and the growth rate of removal efficiency gradually decreased. These observations were similar to the results reported by Chen et al. [23]. After refluxing of nitrified liquid back to the anoxic zone, the organic carbon in the influent was used for denitrification, which improved the TN removal rate. Simultaneously, the COD concentration in the anoxic zone decreased after dilution by refluxed effluent. This led to the alleviated influence on autotrophic microorganisms, which

further enhanced anammox. However, the high reflux ratio not only led to a high nitrate load in the anoxic zone but also reduced the C/N value of the system. Theoretically, nitrite denitrification requires a ratio of COD to total Kjeldahl nitrogen (TKN) greater than 2.5, while nitrate denitrification requires a COD/TKN of 4.0 [24,25]. Therefore, a lack of carbon sources can hinder the process of denitrification. Under the condition of low influent C/N in this experiment, high nitrification reflux ratio could have aggravated the problem of insufficient available organic carbon sources. Therefore, considering energy saving and reduced consumption combined with $NH_4^+$-N and TN effluent concentrations, the optimal reflux ratio was determined as 250%.

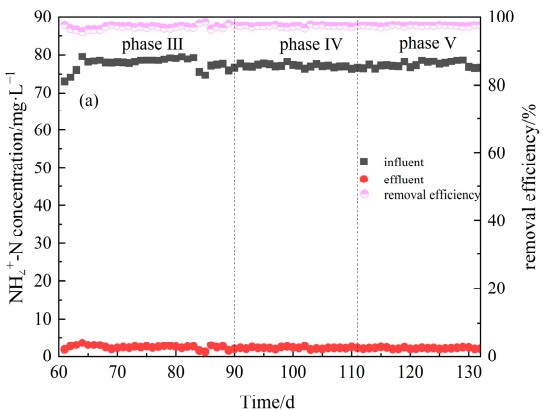 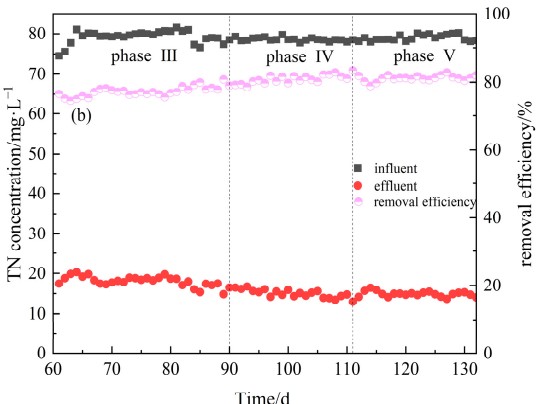

**Figure 5.** Impact of reflux ratio on (**a**) $NH_4^+$-N and (**b**) TN removal efficiencies.

### 3.3. Analysis of Functional Microbial Community for Systemic Nitrogen Transformation

3.3.1. Alpha-Diversity of Microbial Community

The microbial community structure was analyzed in the sludge samples collected from the anaerobic zone, anoxic zone, and aerobic zone (i.e., An, A, and O) in phase V to explore the mechanism of efficient nitrogen removal in the system. The alpha diversity index of the sludge samples is summarized in Table 2.

**Table 2.** Alpha diversity index of sludge samples.

| Sample | Diversity Index | | Richness Index | | Coverage Index/% |
|---|---|---|---|---|---|
| | Shannon | Simpson | Ace | Chao | |
| An | 5.68 | 0.010 | 1742.13 | 1733.28 | 99.93 |
| A | 5.55 | 0.011 | 1573.91 | 1622.82 | 98.83 |
| O | 5.48 | 0.014 | 1523.88 | 1581.04 | 99.41 |

As shown in Table 2, the coverage indices of all samples were greater than 98%, which confirms that the sequencing volumes covered most species in the samples. The Shannon and the Simpson indices showcase the diversity in the microbial community. A high Shannon index and a low Simpson index indicate high microbial diversity in the microbial community. The Chao index and Ace index reflect microbial community richness. From the anaerobic zone to the aerobic zone, a decrease in the alpha diversity index was observed, which indicates the decline in the richness and diversity of the microbial community. The decline in microbial diversity and richness can be due to the low-carbon and low-oxygen environment leading to higher microbial mortality.

3.3.2. Phylum-Level Microbial Community Composition

Figure 6 shows the composition of sludge samples at the phylum level. Phyla with >1% relative abundance were regarded as the main phyla, and the remaining phyla were classified as "others". Chloroflexi, Proteobacteria, Bacteroidetes, Planctomycetes, Acidobacteria,

Verrucomicrobiota, and Armatimonadetes were observed as the top seven phyla in terms of species abundance. The phylum-level microbial composition observed in this study was similar to the composition reported by previous researchers [26,27]. The results implied that the most dominant microbial phyla in the system were Chloroflexi (18.37–30.66%), Proteobacteria (17.31–29.05%), and Bacteroidetes (16.04–21.92%).

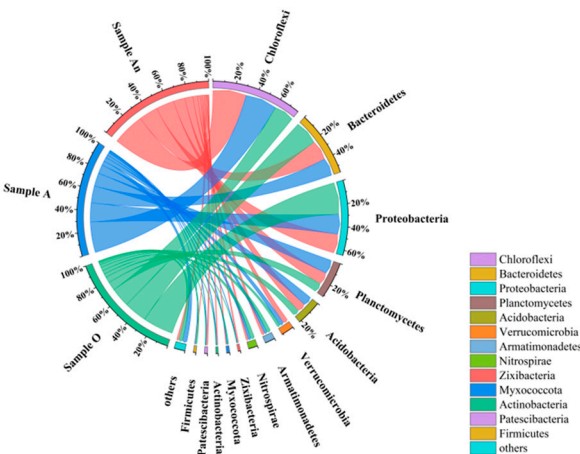

**Figure 6.** Circos diagram representing phylum distribution in the microbial community.

As a common phylum in the anammox system, Chloroflexi consists of mostly linear facultative anaerobic bacteria, which move by gliding. A high abundance of this phylum was also observed in the ammoxidation study carried out by Ya et al. [28]. Some literature has reported that Chloroflexi can contribute to the decomposition of metabolites in AnAOB and reflect the denitrification ability of anammox sludge [29,30]. Furthermore, the large number of filamentous bacteria belonging to the Chloroflexi phylum may serve as skeletons or carriers in the sludge system [31]. As the main phylum involved in the formation of granular sludge, the filamentous bacteria from the Chloroflexi phylum can intertwine to form a granular core, which is conducive to the formation of anammox granular sludge [32]. Therefore, Chloroflexi can be considered a significant phylum in the anammox system. A relevant study revealed that Proteobacteria contained a variety of bacteria associated with denitrification function and could reduce $NO_3^-$-N generated by anammox through endogenous denitrification [33]. Most of the bacteria in the Bacteroidetes phylum are anaerobic rods that could live in anoxic and hypoxic environments, and many bacteria (i.e., AOB, NOB, and denitrifying bacteria) have been reported to be closely associated with nitrogen conversion [34]. Due to uneven aeration, some anoxic areas existed in the aerobic zone of phase V under low DO operating conditions, which may have favored the survival of anaerobic Bacteroidetes. The abundances of Planctomycetes, the main phyla of anammox bacteria, were 10.95%, 12.54%, and 8.09% in the three samples (i.e., An, A, and O), respectively. A similar trend was observed in the anammox study by Xu et al. [35], which reported a higher abundance of Proteobacteria and Bacteroidetes than that of Planctomycetes. Some bacteria leading to short-range denitrification in Bacteroidetes can reduce $NO_3^-$-N and accumulate $NO_2^-$-N in the aerobic zone to provide substrate for anammox bacteria. Therefore, high abundances of Bacteroidetes (21.92%) and Planctomycetes (8.09%) were detected in the aerobic zone.

### 3.3.3. Genus-Level Microbial Community Composition

The genus distribution in the sludge microbial community from different zones is shown in Figure 7. The genus with <0.05% relative abundance were classified as "others" and are not shown in the figure. The dominant AnAOB detected in the $A^2$/O system were *Candidatus Brocadia* (0.22–0.35%), *Pirellula* (0.21–0.33%), and *Gemmata* (0.1–0.21%). Compared to other types of AnAOB in the Brocadiaceae family, *Candidatus Brocadia* has been reported to utilize some organic matter as an electron donor, which helps these

microbes to surpass the substrate competition [36]. Heterotrophic denitrifying bacteria in the anaerobic zones have been reported to inhibit AnAOB at COD $\geq$ 200 mg·L$^{-1}$, with absolute inhibition at COD > 290 mg·L$^{-1}$ [37]. The biodegradable organic carbon source in influent greatly promoted denitrification by heterotrophic bacteria while inhibiting the competition for $NO_2^-$-N substrate with AnAOB. The high COD concentration in the influent was the main reason for the genus-level relative abundance of AnAOB being stable at 0.79% in the system. Chen et al. [38] also found the obvious presence of anammox in the reaction under high COD concentration and low relative abundance (0.76%) of AnAOB.

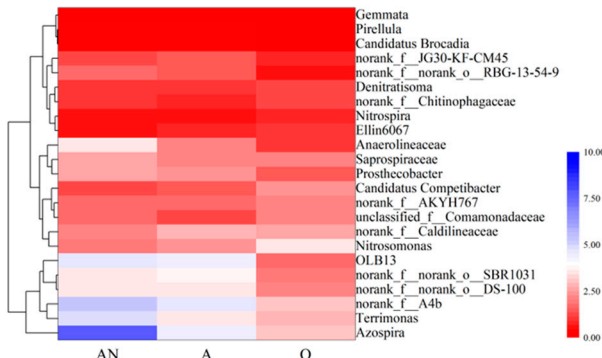

**Figure 7.** Thermal map of microbial community structure at the genus level.

At the same time, other typical microorganisms (i.e., AOB, NOB, and denitrifying bacteria) that contributed greatly to the nitrogen-removal performance of the A$^2$/O system had high relative abundance in the system. The relative abundance of *Nitrosomonas* (AOB) and *Nitrospira* (NOB) in the aerobic zones were 3.47% and 0.81%, respectively, and the abundance of AOB was about 3 times higher than that of NOB. It confirmed the phenomenon of partial nitrification in the aerobic zones of the system at low DO concentrations. Earlier, Bartosch et al. [39] also reported similar findings highlighting the high sensitivity of NOB to DO and weaker competitiveness than AOB under low oxygen conditions in activated sludge. The nitrobacteria *Ellin6067* (0.77%) was also detected in the system. A previous study has reported that *Ellin6067* is involved in the partial nitrification process, and it can jointly build a PN/A microbial system with AnAOB to promote the anammox in the system [40]. *Azospira*, *Terrimonas*, *Anaerolineaceae*, *Prosthecobacter*, and other denitrifiers that have been reported in previous studies [41,42] were also found to be highly prevalent in the anaerobic and anoxic zones of the system. *Denitratisoma* (1.23%) genus was also found in the aerobic zone of the system, which consists of denitrifying bacterial species with the ability to aid AnAOB in removing nitrogen [43]. A 0.89% relative abundance of *Denitratisoma* was detected in the AGS-SBR reactor studied by Xin et al. [44], which was operated in the low-oxygen condition for a long time.

In addition, *norank_f__norank_o__SBR1031*, *norank_f__A4b*, and *OLB13* belonging to the Chloroflexi phylum were also dominant species in the A$^2$/O system. These bacteria were proved to be able to adapt to low-carbon conditions and were active in collaborative cooperation with AOB [45]. In anammox systems, they can not only provide a biofilm framework for AnAOB but also eliminate the inhibitory effects of DO and COD on AnAOB. From the perspective of the genus-level structure, the synergistic actions of various nitrogenization-related bacteria in the reactor resulted in a good nitrogen removal efficiency of the system.

From the results of microbial community composition testing, it can be found that heterotrophic denitrification and anammox (autotrophic) occurred in the anaerobic and hypoxic regions of the reactor. In the aerobic zone, most nitrifying bacteria were AOB (Nitrosomonas), and $NH_4^+$-N was mainly nitrated to $NO_2^-$-N. Due to the inhomogeneity and inadequacy of aeration, a small number of denitrifying bacteria (i.e., Denitratisoma)

also carried out denitrification reactions in the aerobic zone. The mechanism of effective denitrification in the system is shown in Figure 8.

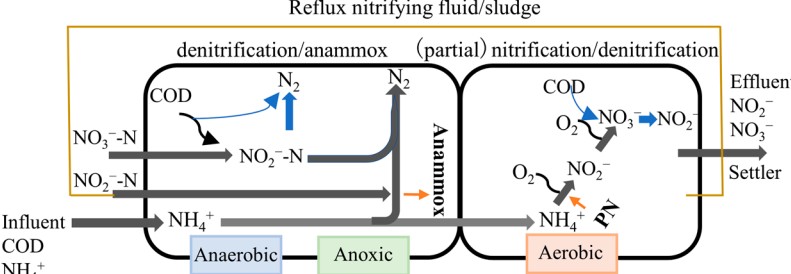

**Figure 8.** Schematic diagram of the mechanism of effective denitrification in the system.

## 4. Conclusions

A continuous $A^2$/O process based on PN/A was implemented in this study for treating wastewater with low C/N (C/N = 3). Effective pollutant removal was obtained with the average $NH_4^+$-N and TN concentration of 2.15 and 14.93 mg·$L^{-1}$ in the effluent, under controlled DO of 0.5–0.8 mg·$L^{-1}$ in each aerobic zone, and the optimum reflux ratio of 250%. In addition, the dominant bacterial phyla in the system were Chloroflexi (26.03%), Proteobacteria (21.48%), and Bacteroidetes (18.65%). At the genus level, a high abundance of *Candidatus Brocadia*, *Pirellula*, and *Gemmata* was detected, and the overall relative abundance of AnAOB was 0.79%. *Nitrosomonas* and *Nitrospira* were the dominant AOB and NOB in the system (aerobic region), with relative abundance of 3.47% and 0.81%, respectively. AnAOB cooperated with various denitrifying microorganisms for nitrogen removal in the PN/A system. Based on these results, the PN/A system constructed with an $A^2$/O process is a promising technology to provide a reference for the low C/N wastewater treatment by WWTPs in China.

**Author Contributions:** Conceptualization, Methodology, Investigation, Formal Analysis, Writing-Original Draft, L.Y.; Data Curation, Writing—Original Draft, Y.Z.; Writing—Review and Editing, L.T. and S.C.; Conceptualization, Resources, Writing—Review and Editing and Funding acquisition, X.Z. All authors have read and agreed to the published version of the manuscript.

**Funding:** This research received no external funding.

**Data Availability Statement:** The data presented in this study are available on request from the corresponding author. The data are not publicly available due to the nature of this research.

**Conflicts of Interest:** The authors declare no conflict of interest.

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
