# Peer review of "Analysis of the Partial Nitrification/Anammox Performance and Microbial Structure of Low C/N Wastewater by A2/O Process"

_water, doi:10.3390/w15122300_

Round 1

Reviewer 1 Report

Nitrogen-removal from low C/N wastewater is important for wastewater treatment using A2/O. It is interesting to improve the N-removal efficiency through partial nitrification/anammox and study the mechanism. Therefore, this manuscript is relevant to the topic of this journal.

The manuscript has focused on optimizing the process parameters to achieve the partial nitrification/anammox system with advanced NH4+-N removal. The experiment has covered sufficient aspects of the system. The results were reasonable. The analysis and explanation of the mechanism generally made sense. I would like to recommend the publication of this study in the Water. Some minor improvements could be considered.

Line 159-160, dilution by refluxed effluent

Line 168-169, this paragraph could be combined with the last one.

Line 200, 214 and so on, anammox bacteria and AnAOB are both used for anaerobic ammonia-oxidizing bacteria. It should be consistent in the context.

Reviewer 2 Report

The authors establish a partial nitrification/anammox system for the anaerobic-anoxic-aerobic treatment of low C/N wastewater, with very good results, which will be important for the effective removal of nitrogen in wastewater treatment plants, not only municipal in China, but in all those with the same problem.

In the document, I included some suggestions. In the references section, there are some citations that do not have the last name of the authors.

Reviewer 3 Report

This study reports that A2/O process can achieve advanced nitrogen removal of low C/N 20

wastewater based on PN/A by optimizing conventional process parameters. This research is well-designed and conducted. The paper is well written. Outcomes of this study may provide practical engineering applications a reference for nitrogen removal based on A2/O process. Therefore, I recommend publication after addressing following issues.

1.      TOC should be measured.

2.      The parameters for the synthetic wastewater with CH3COONa and NH4Cl should be provided.

3.      The Introduction is too simple, it is better to show your thinking.

4.      The advantages, novelty and recent advances of A2/O process in wastewater treatment should be illustrated.

5. The number of repeated experiments should be given since the error bars are shown.

6The advanced techniques for wastewater treatment should be introduced to keep abreast of the latest research trends. e.g.: Chem. Eng. J., 2023, 455, 140943; Adv. Fiber Mater., 2022, 4, 1620, Adv. Fiber Mater., 2022, 4, 736, Adv. Fiber Mater., 2023, 5, 994-1007, these works are informative for readers.

7.      The current treatment results should also be compared with the reported studies.

8.The mechanism of efficient nitrogen removal in the system is suggested to be analyzed and discussed in-depth.

9. In Table 1, TOC values during different periods should be provided.

Minor improvement is suggested

Round 2

Reviewer 3 Report

The revised manuscript is ready for publication.